# Erectile Dysfunction: A Primer for in Office Management

**DOI:** 10.3390/medsci7090090

**Published:** 2019-08-29

**Authors:** Samir Sami, Noah Stern, Andrew Di Pierdomenico, Brandon Katz, Gerald Brock

**Affiliations:** 1Division of Urology, Department of Surgery, Western University, 268 Grosvenor Street, Room B4-663, London, ON N6A 4V2, Canada; 2Medical School, University College Cork, College Road, T12 K8AF Cork, Ireland

**Keywords:** erectile dysfunction, treatment optimization, PDE-5 inhibitors, testosterone

## Abstract

**Introduction:** Optimizing erectile dysfunction (ED) remains a clinically significant endeavor as insufficient outcomes from oral, injectable and even surgical approaches to treatment remain less than ideal. In this report, we integrate evolving knowledge and provide an algorithmic approach for the clinician to fine-tune management. **Methods:** We performed a PubMed and Medline search of Erectile Dysfunction treatment optimization, enhanced patient efficacy for ED, and why men fail ED treatment. All relevant papers for the past two decades were reviewed. **Results:** Establishing the goals and objectives of the patient and partner while providing detailed instructions for treatment can minimize failures and create an environment that allows treatment optimization. A thorough work-up may identify reversible or contributing causes. We identified several areas where treatment of ED could be optimized. These include; management of associated medical conditions, lifestyle improvements, PDE5 inhibitor prescription strategies, management of hypogonadism and the initiation of intracavernosal injection therapy (ICI). **Conclusions:** In our view, once a man presents for help to the clinician, use of the simple strategies identified in this review to optimize the tolerability, safety and effectiveness of the selected treatment should result in enhanced patient and partner satisfaction, with improved outcomes.

## 1. Introduction

The modern era of erectile dysfunction (ED) treatment was ushered into the mainstream consciousness of physicians with the Massachusetts Male Aging Study (MMAS), published in 1994 [1]. It demonstrated that evaluation of ED was possible, that more than 50% of men between 40 and 70 years of age, had some level of sexual dysfunction and identified key risk factors. The past two decades have been witness to dramatic advances in both diagnostic and therapeutic approaches in management of men and their partners suffering from ED. Use of duplex ultrasound imaging which enables the treating physician to evaluate cavernous arterial blood flow, smooth muscle content of the penis and visualize structural abnormalities within the penis, coupled with a greater understanding of the role and importance of testosterone, are just two examples of these developments [2]. Arguably more impactful than these advances has been the cultural change in comfort level among many societies to talk about ED and the agreement by many to seek help. These changes now make it acceptable for many patients to ask for help and have trained clinicians able to evaluate and offer a wide range of therapies, some reversing the underlying pathology, while others provide enhanced penile function.

While there existed effective therapeutic agents prior to the approval of sildenafil in 1998, they were highly invasive and fraught with significant adverse events that limited their widespread acceptance. Intracavernous vasoactive agents, initially popularized in the 1980s and 1990s, required intra-penile injections, exposed the user to the risk of priapism and corporal fibrosis, and could only be offered by specialty centers. Following the approval of sildenafil citrate (Viagra) in 1998, the field has been transformed into one in which the vast majority of caregivers and prescribers are now non-experts. The expansion of the number of prescribers has led to many more patients being treated, but, given the lack of expertise among some treating physicians, has resulted in treatment failures.

Despite the great strides which have been achieved in the field, clinicians are still faced with men who are prescribed ED therapy but either do not return or are disappointed as a consequence of the medical therapy being deemed ineffective. It is the goal of this review to define the optimal strategy that can be used to refine the diagnosis and fine-tune the selection of therapy to provide the user and their partner with the ideal set of instructions and information for use, that result in an effective outcome.

Previous reports have identified a number of key elements of diagnosis and treatment that can increase success rates: [3]
Understanding that not all attempts will be positive—realistic expectations of effectiveness.Men with ED often have underlying medical issues and optimization of these can improve outcome.Reinforcing timing and dosing information for oral agents.Choice of agent needs to meet patient and partner goals.

## 2. Methods

We performed PubMed and Medline searches of ED treatment optimization, enhanced patient efficacy for ED and why men fail ED treatment, reviewing all relevant English language papers for the past two decades.

## 3. Results

### 3.1. Diagnosis and Assessment Options

Critical to the diagnosis and management of ED is patient engagement and a supportive physician/patient relationship [4]. Patient goals and expectations are at the forefront of the discussion and should incorporate the partner when possible [5,6]. Although patients will often present with specific symptoms, emphasis should be placed on evaluating and addressing any underlying conditions [5,7,8].

The ED assessment begins with a comprehensive history and a physical exam. The history should focus on medical, sexual and psychosocial contributors to ED with specific attention to risk factors including cigarette smoking, diabetes, hypertension and cardiovascular health status and a history of pelvic surgery [5,8]. Validated questionnaires such as the International Index for Erectile Function (IIEF), the Sexual Health Inventory for Men (SHIM) and the Erection Hardness Score (EHS) are useful in both assessing ED and monitoring for response to treatment [6,9,10]. Clinicians should aim to ascertain the patient’s concerns, impact of their symptoms and goals of treatment [7].

A focused physical exam should be conducted aiming to identify any genital abnormalities (i.e., penile plaques) that may be contributing to ED or signs of systemic comorbidities such as endocrine, sensory loss or evident vascular insufficiency [5].

Consensus guidelines recommend laboratory testing assessing for common etiologies of ED including diabetes, dyslipidemia and atherosclerosis [5,8]. Hormone testing is an option for further assessment, though its role remains controversial. Most societies agree it has a role in men who have failed PDE-5 inhibitors, exhibit hypoactive desire, and in all men with both ED and diabetes due to the high concomitance of hypogonadism, ED, and diabetes [5,11]. Further testing including assessment of thyroid function and a comprehensive hormone panel may be indicated in select patients [5,9].

Specialized testing such as a psychological assessment, nocturnal penile tumescence, vascular assessment, and penile duplex ultrasound (DUS) may aid in further clarifying the etiology of ED; however, it is often reserved for select cases. DUS is often combined with intracavernosal injection (ICI) of vasoactive substances allowing for the demonstration of penile plaques, curvature and vascular response. In men who have failed PDE-5 inhibitors, it may serve as a useful adjunct for diagnosing ED of vasculogenic etiology [8].

### 3.2. Associated Conditions

Multiple comorbidities have well-established connections to ED including diabetes mellitus, hypertension, dyslipidemia, metabolic syndrome, depression, and lower-urinary-tract symptoms [5,7,8,12,13,14]. Given the association between ED and cardiovascular, coronary artery, and cerebrovascular disease, many societies recommend further evaluation of men presenting with arteriogenic ED as it may be a harbinger for preexisting or subclinical disease [5,8,15]. Modifying these risk factors through lifestyle changes and pharmacotherapy may improve, prevent or, in some cases, reverse ED entirely and lead to improved overall health.

Patients with cardiac conditions that pose a significant health risk should be assessed by a cardiologist prior to commencing PDE-5 inhibitor use. Visual conditions that also necessitate caution when considering PDE-5 inhibitor use are non-arteritic anterior ischemic optic neuropathy and a crowded optic disc [5,7,8].

Gupta et al. (2011) performed a meta-analysis to assess the effect of both lifestyle interventions and pharmacotherapy (atorvastatin) on erectile function [16]. The results showed an IIEF score improvement of 2.7 points. Further analysis separating lifestyle and pharmacotherapy identified an IIEF score improvement of 2.4 and 3.1, respectively. While the clinical significance is questionable, given PDE-5 inhibitor IIEF improvements of up to 7–10 points, it is important to note that three of the six trials and 35% of the total patients involved had concurrent PDE-5 inhibitor use. Thus, CVD risk factor reduction through lifestyle and pharmacotherapy may provide additive benefit with PDE-5 inhibitor use for erectile function.

Esposito et al. (2004) identified a relationship between physical activity and erectile function [14]. A total of 110 obese men with ED and minimal comorbidity were randomized to a weight-loss/increased physical activity program vs. receipt of general health information (control). The groups were followed for two years. Improvements were seen in body mass index (BMI) and IIEF score conferring a four-point increase compared to the control arm. In another trial by Esposito et al. (2006), 65 men with metabolic syndrome were randomized to a Mediterranean diet vs. a standard diet [17]. ED was not a specific inclusion criterion. Following two years, improvements in endothelial function and inflammatory markers were noted in the Mediterranean diet group. This was associated with a 3.7 IIEF score improvement in the Mediterranean diet group. Although these studies used small sample sizes with limited generalizability, they provided evidence as to the effect of lifestyle interventions on erectile function.

### 3.3. Psychological Disorders

The three major psychological aspects that are most relevant to sexual function are: (1) depressive illness, (2) performance anxiety, and (3) couple dysfunction. These can be explored within a few minutes during consultation, if indications of significant distress are found, the patient can be referred appropriately [5,7,8,18].

Depression is an independent risk factor for ED. Furthermore, certain antidepressants may exacerbate or lead to ED. PDE-5 inhibitor therapy in patients suffering from depression can enhance erectile function even with concurrent antidepressant medication use, for example, selective-serotonin reuptake inhibitors (SSRIs) [19].

### 3.4. Medical Management and Optimization Strategies

PDE-5 inhibitors promote the erectile response by blocking the degradation of cyclic guanosine monophosphate (cGMP). This promotes dilation of the corpus cavernosal smooth muscle and augments erection following sexual stimulation [20].

Commercially available PDE-5 inhibitors include sildenafil, vardenafil, tadalafil, and avanafil. Although no specific RCTs have been conducted comparing each of these agents, meta-analysis indicates no significant differences in the efficacy of PDE-5 inhibitors relative to each other. Adverse events, such as dyspepsia, headaches, nasal congestion, back pain, and visual disturbances, are comparable with subtle differences [3,4,5,8,19,20,21,22].

In patients with ED, sildenafil was noted to improve erections in up to 70% [22]. The efficacy of this treatment combined with its favourable adverse event profile have made it a first-line medical therapy for ED. Sildenafil can be prescribed at a standard dose of 50 mg and titrated according to response and side effects. Maximum plasma concentration is reached at 60 min post-consumption, and the therapeutic effect lasts for 12 h, making it best suited to those individuals whose sexual encounters occur at a predictable time [23,24]. However, in our opinion, a starting dose of 100 mg has better efficacy and a tolerable side effect profile and reduces the risk of nonresponse. It is our opinion that this should be the starting dose.

Tadalafil is an alternative for those patients who may benefit from prolonged plasma drug concentration over a 36 h period. It is useful in patients who desire that sexual activity occurs more spontaneously [25,26]. It is effective within 30 min of administration. It is available in dosing increments spanning 2.5–20 mg. In our opinion, a 20 mg starting dose has a similar side effect profile but has greater efficacy across broad populations and, as such, we recommend that it be the starting dose.

Vardenafil, like tadalafil, achieves effective concentration 30 min after administration. Unlike sildenafil, its effect is diminished by fatty meals [27,28]. Patients are typically started on a 10 mg starting dose. Our recommendation is that a 20 mg dose be the optimal starting dosage for reasons of efficacy and tolerability.

Avanafil, the most recent PDE5i to gain approval in North America, is highly efficacious at inhibiting the PDE-5 enzyme. This makes optimizing treatment with regards to a diminished side effect profile less challenging. Patients are regularly advised to take a starting dose of 100 mg 15 to 30 min prior to being sexually active. Patients are advised to take the drug once per day and to avoid taking the drug with food for maximum effect [29,30].

Pharmacokinetic differences between the agents may provide physicians with the opportunity to individualize treatment. For instance, the longer half-life of tadalafil can potentiate erectile responses up to 36 h and its low-dose formulations allow for daily usage [31]. A shorter half-life, however, may provide more rapid reversibility of the agent and minimize adverse events [21,22,29,32].

### 3.5. Contraindications and Drug–Drug Interactions with PDE-5Is

Caution should be used when PDE-5i are co-administered with certain medications, such as nitrates or guanylate cyclase stimulators (these are typically contraindicated). Hypotension may also occur with concurrent use of alpha-blockers and certain antiretroviral drugs. In this latter scenario, we suggest thorough patient counselling, dose titration and temporal separation of the intake of the different medications by several hours [5,7,8].

### 3.6. Management of Erectile Dysfunction in Patients with Spinal Cord Lesions

A systematic review and meta-analysis evaluating PDE-5 inhibitor use noted a marked improvement of erectile function in patients with spinal cord injuries. The level of spinal cord injury appears to be a predictor of PDE-5 inhibitor success, wherein lower motor neuron lesions may have decreased efficacy [33,34]. Motivated men who respond inadequately to PDE-5 inhibitors may benefit from specialist referral for further management.

### 3.7. Prostate Cancer

ED in men with prostate cancer is often multifactorial and is a common side effect amongst all treatment modalities [35].

Post-prostatectomy, men may experience ED due to nerve injury despite nerve-sparing surgical approaches. This can contribute to the apoptosis of cavernosal smooth muscle and contribute to persistent post-operative ED [36]. Erectile function recovery can take up to four years post-operatively [8]. In vitro and in vivo data suggests PDE-5 inhibitor therapy may improve post-operative tissue oxygenation and thereby inhibit hypoxia-associated fibrosis of the cavernosal smooth muscle [36]. In a small trial, early use of sildenafil versus a placebo was seen to preserve smooth muscle content [37]. Further studies have demonstrated that post prostatectomy patients taking sildenafil nightly experience earlier return of spontaneous erections [6]. Favourable prognostic factors for response to PDE-5 inhibitor therapy include preoperative erectile function, age <65 years, and degree of neurovascular bundle preservation [38]. Penile rehabilitation strategies intended to enhance erectile function recovery using various PDE-5 inhibitors (tadalafil, sildenafil, and vardenafil) have been examined in a randomized placebo-controlled fashion. All three PDE-5 inhibitors improved erectile function with use [39,40,41]. Only one study (Padma-Nathan et al., 2008) noted nightly sildenafil for nine months followed by a drug-free washout period led to a statistically significant return of spontaneous erections [39]. In contrary, a study by Pavlovich et al. in 2013, noted no difference in recovery of erectile function, as measured by nocturnal penile tumescence and IIEF scores, when comparing nightly sildenafil versus on-demand dosing [42]. Shortcomings of the data are length of follow-up (up to 13 months), large dropout rates, and heterogeneity in operative technique and surgical outcomes. The role of PDE-5 inhibitors in penile rehabilitation remains to be established [43].

Incidence of ED amongst patients undergoing external-beam radiotherapy generally ranges from 30–40% and this effect is initially delayed—occurring after the first year. Brachytherapy may help limit the detrimental effects from radiation through targeted mapping and seed implantation. Regardless of modality, radiotherapy-induced ED appears to be arteriogenic with no clear relationship between radiation dose and the subsequent development of ED. PDE-5 inhibitors remain first-line agents for medical therapy. Similarly, for post-prostatectomy men, there may be a role for early initiation of penile rehabilitation initiatives post radiotherapy [44].

Androgen deprivation therapy (ADT) is critical in the management of metastatic prostate cancer and can be used in a neoadjuvant or adjuvant setting. Androgen depletion causes a multitude of changes leading to reduced sexual desire. This includes, and is not limited to, reduction in response to endogenous vasodilators, altering compliance of corpus cavernosum and resulting in dysfunction of the veno-occlusive mechanism important in erectile function [45]. Intermittent ADT, as opposed to continuous ADT, may be used in the right clinical setting and this may help alleviate sexual dysfunction [46]. The treatment algorithm for such patients remains similar with conservative and medical treatment options trialed initially; however, early specialist referral may be warranted.

### 3.8. Treatment Failure

Although PDE-5 inhibitors have transformed the treatment paradigm for ED, they are not suitable for all patients and not efficacious for others. When addressing men who have failed a trial of PDE-5 inhibitors, ensuring a proper dosing technique is essential. In a sildenafil treatment trial amongst 253 non-responders, 40% of patients were able to achieve a long-term satisfactory response through in-office education and evaluation. The mean office time was 12 min, suggesting a short visit can improve efficacy prior to declaring true treatment failure [47]. Additional strategies for suboptimal responders include dose escalation or employing a low-dose daily dosing strategy [7]. Switching between different types of PDE-5 inhibitors does not appear to improve response rates [7].

Hypertension, dyslipidemia, tobacco use, and poor glycemic control have been identified as independent predictors of poor responses to sildenafil [19]. In patients deemed unresponsive to sildenafil, two placebo-controlled trials showed treatment of hypertension and dyslipidemia with an angiotensin-converting enzyme and a statin, respectively, improved erectile function [3,4,6]. Studies examining poor glycemic control have shown that increased hemoglobin A1c levels were correlated with an increased incidence of ED over time [48]. However, it is not clear whether intensive glycemic control in men with elevated hemoglobin A1c levels leads to improved erectile function. Although not directly compared, the response rates as expected to PDE-5 inhibitor therapy are less in men with diabetes compared to non-diabetic men. No particular oral agent appears to be superior. Oral PDE-5 inhibitors remain first-line in this patient cohort with no significant increase in adverse events [13,14,48].

### 3.9. Hypogondal Men and Testosterone Supplementation

Androgens act at the central nervous system, peripheral nervous system and in the penis to preserve erectile function. At the penis, these actions include maintenance of trabecular smooth muscle, regulation of precursor cell differentiation in the corpus cavernosum, maintenance of endothelial function and maintenance of penile connective tissue including the tunica albuginea [49,50,51]. In animal models, histologic changes coincident with ED can be induced by castration and then reversed by testosterone therapy (TT) [52,53]. Limited data suggests similar histologic changes occur in humans and may be related to ED [54,55].

Hypogonadism, compared to castration, may be a diagnostic challenge owing to variation in testosterone levels and responsiveness to androgens in the population. Threshold levels at which ED may present have been proposed by some authors, for example, 8.5 nmol/L for isolated ED and 10.4 nmol/L for two of the following three symptoms: ED, decreased morning erections or low libido [56,57]. These values align with the lower limit of normal of most testosterone assays and the cut-off for hypogonadism in many clinical guidelines [58]. The suggestion is that men with low testosterone and ED may benefit from replacement.

TT has been utilized as monotherapy for treatment of ED. One meta-analysis of 14 RCTs that compared the effect of TT to placebo in men with ED demonstrated improved IIEF scores [59]. The greatest effect was observed when the analysis was restricted to those participants with a baseline testosterone <8 nmol/L who demonstrated a mean increase in IIEF-EFD score of 2.95 points [59]. A similar increase of 2.64 points on the IIEF-EFD was seen in a recent, large RCT of hypogonadal men with sexual symptoms treated with testosterone gel [60]. A more recent meta-analysis restricted to four RCTs including hypogonadal men also demonstrated efficacy of TT in improving erectile function in this population [61].

In hypogonadal men, androgens may improve the response to PDE5i in previous non-responders, though the mechanisms of synergy are unclear [62]. One study of sildenafil non-responders with a serum testosterone in the lower quartile of the normal range (10–13 nmol/L) demonstrated a 7.4-point improvement in the IIEF-EFD after treatment with combined testosterone and sildenafil vs. sildenafil alone [63]. In comparison, a meta-analysis of 12 trials evaluating the effect of TT added to PDE5i therapy demonstrated no significant effect when the analysis was limited to the five placebo-controlled trials [64]. A limitation in these trials has been a potential direct effect of PDE5i on serum testosterone levels, negating the additive benefit of TT [65].

### 3.10. History of ICI

After first being introduced in 1982, ICI therapy has provided robust relief for men with neurogenic, vasculogenic, psychogenic and mixed erectile dysfunction with success rates up to 95% in properly identified populations [66,67,68]. Intracavernous cocktails can be delivered as mono-agents or as combinations utilizing agents that address multiple molecular targets in the erectile pathway.

### 3.11. Alprostadil

Currently available as a monoagent and as a component of Trimix and Quadmix, Alprostadil is a synthetic prostaglandin E1 analogue. Prostaglandin E1 exerts pro-erectile effects by increasing intracellular cAMP by activating adenylate cyclase, resulting in smooth muscle relaxation [69]. Alprostadil exhibits a dose response with dosages titrated up from 1.25 to 20 μg with patient satisfaction rates approaching 80% [70]. It is currently the only FDA-approved injectable monoagent in the treatment of ED. The most common adverse events are related to site of injection including pain (11%) and hematoma (1.5%). As with other injection therapy, alprostadil carries the risk of priapism, fibrosis and pain, though the rate of priapism is the lowest (0.36%) among available agents [71]. Our recommendation would be that a starting dose of 5 µg of PGE-1 and titration up to 20 µg would salvage most non-responders and optimize therapy. If pain is a limiting factor, a switch to Bimix or Papaverine montherapy can provide pain-free injection therapy. Care should be taken in neurogenic patients where a robust response is expected given their normal underlying vasculature; therefore, a smaller starting dose (2.5 µg) and a gentle titration may be appropriate.

### 3.12. Papaverine

Papaverine acts as a nonselective PDE-5 inhibitor preventing the degradation of cAMP and cGMP. It has been shown to both increase the arterial flow and decrease venous outflow, with both mechanisms working synergistically to increase tumescence [72]. Despite being the first intracavernosal agent, described by Virag in 1982, it shows poor overall efficacy compared to other agents and a higher side effect profile (most notably fibrosis) [66]. As a result, papaverine is seldomly used as monotherapy, but is often present in formulations [73,74].

### 3.13. Phentolamine

Phentolamine acts as a nonselective alpha-adrenergic receptor antagonist that blocks sympathetic (and thus anti-erectile) stimulation of the corporeal smooth muscle to improve tumescence. It shows limited efficacy as a monoagent [75].

### 3.14. Combination Therapy

Papaverine/phentolamine (Bimix) utilizes the synergistic PDE-5 inhibition of papaverine and alpha blockage of phentolamine, resulting in a response rate of 68.5% [73]. Overall, Bimix appears to have improved success in achieving erections sufficient for penetration compared to papaverine alone (54% vs. 50%); however, a higher incidence of penile fibrosis is observed [76]. The addition of alprostadil to Bimix results in Trimix, and the addition of atropine results in Quadmix. Currently, there is no commercially available formulation of Trimix, with multiple formulations discussed in the literature making direct comparisons difficult. The only prospective trial to date had men progress through four protocols from Bimix to Quadmix in increasing dosages. Of the 625 men involved, all but 15 achieved erections sufficient for penetration [77].

## 4. Conclusions

The ability to restore sexual functioning to the vast majority of men with ED is a present-day reality, owing to advances in diagnosis and therapeutics. In truth, many men will never seek medical care as they have decreased interest, an unwilling partner, or simply believe effective therapy does not exist. In our view, once a man presents for help to the clinician, use of the simple step-wise strategies identified in this review to optimize the tolerability, safety, and effectiveness of the selected treatment should result in enhanced patient and partner satisfaction and improved outcomes (Figure 1).

Use of higher, rather than lower doses, multiple attempts, having the patient and partner understand the time course of the drugs’ absorption, distribution and clearance, coupled with clear detailed instructions for use often can avoid treatment failures and optimize erectile dysfunction therapies.

## Figures and Tables

**Figure 1 medsci-07-00090-f001:**
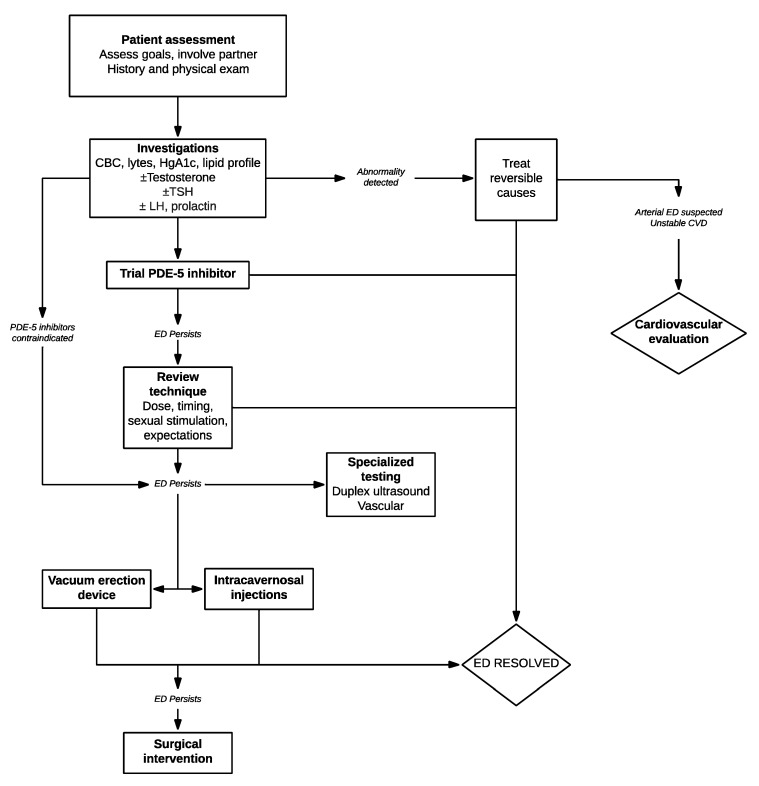
Algorithm for evaluation and management of patients presenting with erectile dysfunction (ED).

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
