# Peer review of "Erectile Dysfunction: A Primer for in Office Management"

_medsci, 2019, doi:10.3390/medsci7090090_

Round 1
Reviewer 1 Report
Comments on the manuscript:
“Erectile dysfunction: a primer for in office management”
This manuscript is an interesting and useful review about the erectile dysfunction (EC) and the effects of various remedies used to restore normal physiology. The authors emphasize that the results are not always satisfactory and propose a medical approach that takes into account the different factors that can lead to an EC.
The manuscript is well written and the data useful. I have only some minor remarks in rde to improve the manuscript.
Line 220: reference Padma-Nathan et al 2008: what is the number of the reference? 37?
Line 257::histological (instead histologic)
Line 313: reference Virag et al 1982: What is the refereence number? Is-it 61? If it is 61, Virag is the single author.
Line 334, figure 1: this algorithm needs to be presented in the text in which the figure will be called. This algorithm could be given in the conclusions.
Line 404: write: phosphodiesterase.
Line 519, reference 61: use lowercases.
Line 529, reference 65: use lowercases.
Author Response
thank you for the review and comments. please see the attached file. included is comments from al three reviewers and a revised version of manuscript
Review Report #1
Line 220: reference Padma-Nathan et al 2008: what is the number of the reference? 37?
Added note to reference 39
Line 313: reference Virag et al 1982: What is the reference number? Is-it 61? If it is 61, Virag is the single author. Line 404: write: phosphodiesterase. Line 519, reference 61: use lowercases. Line 529, reference 65: use lowercases. Line 257::histological (instead histologic)
Corrected
Line 334, figure 1: this algorithm needs to be presented in the text in which the figure will be called. This algorithm could be given in the conclusions.
Algorithm has now been referenced appropriately in the conclusions

Reviewer 2 Report
This review article introduced a primer of erectile dysfunction (ED) practice for general physicians. Though the contents of this review are not so novel for specialists, it seems to be significant for general physicians.
(Minor)
Line 62 to 64. The authors described some patients returned disappointed for ED treatments. I am afraid that several patients have never returned to clinicians because they disappointed and gave up ED treatments. Please add this situation at ED practice. Line 87 to 95. The authors mentioned both subjective and objective ED assessment in this paragraph. I would like to put in erectile hardness score (EHS) to evaluate penile stiffness. Line 96 to 102. In this paragraph, common etiology of ED was described. I recommend adding several risk factors simultaneously. In line 239, tobacco was written. In addition, aging, intra-pelvic operation etc should be described to pay attention in ED practice. Line 206. The relationship between prostate cancer treatment and ED is a very important issue. The authors mainly mentioned about post prostatectomy and ED. Please make some comments for radiation and hormonal therapy. Line 290. The authors described ICI has provided robust relief for men with neurogenic, vasculogenic~. General speaking, ICI induces relaxation penile endothelial cells, and treat ED. Thus, insufficient blood flow to penis results in failure of ED treatment with ICI. I think it is better to amend or delete vasculogenic. Line 304 to 306. The authors recommended that a starting dose of 5ugof PGE-1. When ED was caused after intra-pelvic surgery, ICI should start with low dosage such as 5ug to avoid painful erection. Please make additional comments in this paragraph.
Author Response
Thank you for the review. please see attached file. attached is revised version of manuscript including comments from all three reviewers.
Review Report #2
Line 62 to 64. The authors described some patients returned disappointed for ED treatments. I am afraid that several patients have never returned to clinicians because they disappointed and gave up ED treatments. Please add this situation at ED practice.
. We agree with this point. We have amended the reference to disappointed patients and included the statement that many patients who are truly disappointed may choose not to return to the clinician and as such, this is another important rationale to optimize choice of management at the outset. We have revised the section to make note of this
“Despite the great strides which have been achieved in the field, clinicians are still faced with men who are prescribed ED therapy but either do not return or are disappointed as a consequence of the medical therapy being deemed ineffective.”
Line 87 to 95. The authors mentioned both subjective and objective ED assessment in this paragraph. I would like to put in erectile hardness score (EHS) to evaluate penile stiffness.
Added reference to EHS
Line 96 to 102. In this paragraph, common etiology of ED was described. I recommend adding several risk factors simultaneously.
Added reference to ED risk factors
In line 239, tobacco was written. In addition, aging, intra-pelvic operation etc should be described to pay attention in ED practice.
Now referenced in ED risk factors
Line 206. The relationship between prostate cancer treatment and ED is a very important issue. The authors mainly mentioned about post prostatectomy and ED. Please make some comments for radiation and hormonal therapy.
Added section discussion radiation and androgen deprivation therapy
Line 290. The authors described ICI has provided robust relief for men with neurogenic, vasculogenic~. General speaking, ICI induces relaxation penile endothelial cells, and treat ED. Thus, insufficient blood flow to penis results in failure of ED treatment with ICI. I think it is better to amend or delete vasculogenic.
When administering ICI in vascuologenic patients clinicians certainly expect a lower degree of success, however given the invasiveness of subsequent treatment options the practice at our centre has been to trial ICI in these patients.. The vasculogenic patient often requires higher doses of ICI agents but in our experience the vast majority of cases can be effectively managed with ICI.
Line 304 to 306. The authors recommended that a starting dose of 5ugof PGE-1. When ED was caused after intra-pelvic surgery, ICI should start with low dosage such as 5ug to avoid painful erection. Please make additional comments in this paragraph.
Comment added

Reviewer 3 Report
The topic of this narrative review is of great interest for urologists, general practitioners, oncologists.
The review is well written, but I believe that there are some points that should be discussed. In particular, I suggest to
1) discuss the potential use of flavonoids such as quercetin or isoquercetin as an adjunct to pharmacologic treatment. Please consider:
Am J Clin Nutr. 2016 Feb;103(2):534-41. doi: 10.3945/ajcn.115.122010. Epub 2016 Jan 13.Dietary flavonoid intake and incidence of erectile dysfunction. Cassidy A1, Franz M2, Rimm EB3. Int J Clin Exp Med. 2015 May 15;8(5):7599-605. eCollection 2015. Effects of quercetin on intracavernous pressure and expression of nitrogen synthase isoforms in arterial erectile dysfunction rat model.Zhang Y1, Huang C2, Liu S3, Bai J4, Fan X5, Guo J6, Jia Y7, Zhang Z8, Chen X8, Jia Y8, Zhang P4, Wang B9, Zhang X10. 2) discuss ED differences in prostate cancer patients treated with prostatectomy vs. radiotherapy vs. hormonal therapy 3) discuss alprostadil cream (trademark VITAROS)lease be aware that in the abstract
"We identified several areas where treatment of ED could be optimized, these 24 include; management of associated medical conditions, lifestyle improvements, PDE5 25 inhibitor prescription strategies, management of hypogonadism and the initiation of 26 intracavernosal injection therapy (ICI)."
should read
We identified several areas where treatment of ED could be optimized. These 24 include the following: management of associated medical conditions, lifestyle improvements, PDE5 25 inhibitor prescription strategies, management of hypogonadism and the initiation of 26 intracavernosal injection therapy (ICI).
Author Response
Thank you for the review.
Review report #3
1) discuss the potential use of flavonoids such as quercetin or isoquercetin as an adjunct to pharmacologic treatment
While the use of flavonoids in the treatment of erectile dysfunction is certainly a promising area that warrants further investigation, we feel the data is not sufficiently robust to discuss in this paper given the word limitations we were given.
2) discuss ED differences in prostate cancer patients treated with prostatectomy vs. radiotherapy vs. hormonal therapy
Reference added to radiation and hormonal therapies
3) discuss alprostadil cream
Topical therapies do show efficacy, however their significant side effect profile and high rate of discontinuation by patients have resulted in poor results in practice. We have therefore opted to omit these therapies from this paper as we hope to provide a general algorithm with the highest potential for success. Additionally, their availability is not widespread.
Round 2
Reviewer 3 Report
the authors fail to recognize the value of the proposed topics of discussion that would have surely improved the paper.
I strongly Believe that nutritional supplementation and topical routes of administration are very important in clinical practice.